# Answers to the Frequently Asked Questions Regarding Horse Feeding and Management Practices to Reduce the Risk of Atypical Myopathy

**DOI:** 10.3390/ani10020365

**Published:** 2020-02-24

**Authors:** Dominique-Marie Votion, Anne-Christine François, Caroline Kruse, Benoit Renaud, Arnaud Farinelle, Marie-Catherine Bouquieaux, Christel Marcillaud-Pitel, Pascal Gustin

**Affiliations:** 1Equine Pole, Fundamental and Applied Research for Animals & Health (FARAH), Faculty of Veterinary Medicine, University of Lieège, 4000 Liège 1 (Sart Tilman), Belgium; dominique.votion@uliege.be (D.-M.V.); mcbouquieaux@uliege.be (M.-C.B.); 2Department of Functional Sciences, Faculty of Veterinary Medicine, Pharmacology and Toxicology, Fundamental and Applied Research for Animals & Health (FARAH), University of Liège, 4000 Liège 1 (Sart Tilman), Belgium; benoit.renaud@uliege.be (B.R.); p.gustin@uliege.be (P.G.); 3Department of Functional Sciences, Faculty of Veterinary Medicine, Physiology and Sport Medicine, Fundamental and Applied Research for Animals & Health (FARAH), University of Liège, 4000 Liège 1 (Sart Tilman), Belgium; caroline.kruse@uliege.be; 4Fourrages Mieux asbl, 6900 Marloie, Belgium; farinelle@fourragesmieux.be; 5Réseau d’Epidémio-Surveillance en Pathologie Équine (RESPE), 14280 Saint-Contest, France; c.marcillaud-pitel@respe.net

**Keywords:** equine atypical myopathy, *Acer* spp., risk factors, environment

## Abstract

**Simple Summary:**

Equine atypical myopathy is a severe intoxication of grazing equids resulting from the ingestion of samaras or seedlings of trees from the *Acer* species. The sycamore maple (*Acer pseudoplatanus*) is involved in European cases whereas the box elder (*Acer negundo*) is recognized as the cause of this seasonal pasture myopathy in the Unites States of America. In Europe, young and inactive animals with a thin to normal body condition and no feed supplementation, except for hay in autumn, are at higher risk. The risk is also associated with full time pasturing in a humid environment. Indeed, dead leaves piling up in autumn as well as, the presence of trees and/or woods presumably exposes the horses to the sycamore maple. This manuscript answers the most frequently asked questions arising from the equine field about feeding and management of equines to reduce the risk of atypical myopathy. All answers are based on data collected from 2006 to 2019 by the “Atypical Myopathy Alert Group” (AMAG, Belgium) and the “Réseau d’épidémiosurveillance en Pathologie équine” (RESPE, France) as well as on a review of the most recent literature.

**Abstract:**

In 2014, atypical myopathy (AM) was linked to *Acer pseudoplatanus* (sycamore maple) in Europe. The emergence of this seasonal intoxication caused by a native tree has raised many questions. This manuscript aims at answering the five most frequently asked questions (FAQs) regarding (1) identification of toxic trees; reduction of risk at the level of (2) pastures and (3) equids; (4) the risk associated with pastures with sycamores that have always been used without horses being poisoned and (5) the length of the risk periods. Answers were found in a literature review and data gathered by AM surveillance networks. A guide is offered to differentiate common maple trees (FAQ1). In order to reduce the risk of AM at pasture level: Avoid humid pastures; permanent pasturing; spreading of manure for pasture with sycamores in the vicinity and avoid sycamore maple trees around pasture (FAQ2). To reduce the risk of AM at horse level: Reduce pasturing time according to weather conditions and to less than six hours a day during risk periods for horses on risk pasture; provide supplementary feeds including toxin-free forage; water from the distribution network; vitamins and a salt block (FAQ3). All pastures with a sycamore tree in the vicinity are at risk (FAQ4). Ninety-four percent of cases occur over two 3-month periods, starting in October and in March, for cases resulting from seeds and seedlings ingestion, respectively (FAQ5).

## 1. Introduction

Equine atypical myopathy (AM) is a severe pasture-associated intoxication that may occur in autumn and spring following the ingestion of certain species of maple (*Acer*) seeds and seedlings, respectively. This environmental intoxication is linked to *Acer pseudoplatanus* (sycamore maple) in Europe and *Acer negundo* (box elder) in the US [1,2]. These trees may contain several toxins [3]. The ingestion of samaras or seedlings of the incriminated trees goes with the ingestion of two cyclopropylamino acids, hypoglycin A (HGA) and methylenecyclopropylglycine (MCPG) [4]. These toxins have been confirmed to be implicated in European AM cases. Long before the discovery of the cause of AM, Fowden and Pratt (1973) [3], reported the presence of cyclopropyl derivates in seeds of the different representatives of the *Acer*’s species. Both *Acer pseudoplatanus* and *Acer negundo* seeds have been found to contain HGA and MCPG. On the contrary, other maple trees commonly found in Europe as *Acer platanoides* (Norway maple) and *Acer campestre* (field maple) tested negative for these compounds [3].

In fact, HGA and MCPG are not toxic *per se* but need to be converted into their active metabolites, i.e., methylenecyclopropylacetyl-CoA (MCPA-CoA) and methylenecyclopropylformyl-CoA (MCPF-CoA), respectively [4,5]. Both MCPA-CoA and MCPF-CoA inhibit enzymes that participate in β-oxidation and thus energy production from lipid metabolism [5,6]. The typical sign of intoxication is an acute rhabdomyolysis syndrome unrelated to exercise. This clinical picture may be seen on several horses within a group [7,8,9,10]. In more than 50% of the cases, the following clinical signs were observed: Weakness, recumbency, myoglobinuria, full bladder, stiffness, depression, muscle tremors or fasciculation, reluctance to move, sweating, normothermia, and congested mucous membranes [10,11,12] Atypical myopathy has a high mortality rate (i.e., 74%) that varies between countries and years (from 43% [11] to 97% [13]). The overall mortality rate of 74% average data among countries included in the study of van Galen et al., (2012) and does not take into account the different sources of variability [11]. For example, a study reports a lower mortality rate (i.e., 44%) in hospitalized animals [14]. It is hypothesized that less critical cases are driven to a hospital where appropriate symptomatic treatment is easier to provide. These two factors may contribute to the higher survival rate in an equine hospital than in the field. In any case, only the administration of vitamins and antioxidants has proven to be beneficial for survival [12,15].

In 2004, an alert group named “Atypical Myopathy Alert Group” (AMAG) was launched to warn horse practitioners and owners of the risk peaks. The alerts are released following case declarations and the AMAG regularly updates its data with the latest number of cases. Additionally, to its disease surveillance role, AMAG collects epidemiologic data about AM that has emerged in several European countries since 2006. French cases are gathered in close collaboration with the Réseau d’Épidémio-Surveillance en Pathologie Équine (RESPE) which monitors equine diseases through a network of French sentinel practitioners. Thanks to the European surveillance network, we now know that several Père David’s deer (*Elaphurus davidianus*) have succumbed from this intoxication in different zoos in Germany. This indicates that ruminants pasturing in the vicinity of sycamore trees may also be intoxicated [16].

In light of the high mortality rate and the absence of specific treatment, prevention is the key to avoid intoxication of animals. Before discovering the cause of AM, epidemiological studies revealed risk factors associated with management practices of horse and pasture [15,17] In 2014, the cause of AM was discovered and linked to *Acer pseudoplatanus* in Europe [1]. Despite the cause of the intoxication being known, outbreaks have continued to occur. The development of a condition caused by a native tree has raised many questions among all actors attached to the equine sector.

The most frequently asked question (FAQ) concerns the identification of toxic trees. Other commonly-asked questions involve the feeding and management practice of equids in order to reduce the risk of intoxication. These main FAQs can be summarized as being: (FAQ1) “Which maples are toxic? Is this tree a maple and if so, is it toxic?”; (FAQ2) “How can AM be prevented (at pasture level)?”; (FAQ3) “How can AM be prevented (at horse level)?”; (FAQ4) “Our pasture is surrounded by sycamore maple trees, but no case of AM ever occurred in our grazing horses. Does this mean the pasture is safe for our animals?” and (FAQ5), “When does the risk of AM start and stop in autumn and spring?”.

This manuscript answers to these FAQs regarding horse nutrition and management practices in order to prevent AM both by reviewing the most recent literature and by analyzing epidemiological data gathered since 2006.

## 2. Materials and Methods

### 2.1. Literature Review

A systematic review was performed using on the electronic databases PubMed and Scopus with « atypical myopathy » AND « horse » as keywords. In addition, abstracts of proceedings of meetings dedicated to horses were consulted (e.g., AAEP—American Association of Equine Practitioners Annual Convention, AVEF—Association des Vétérinaires Équins Français, BEPS—Belgian Equine Practitioners Society Study Days, BEVA—British Equine Veterinary Association and, WEVA—World Equine Veterinary Association Congress).

### 2.2. Epidemiological Data

Information regarding AM cases in Europe over a 13-year period (2006–2019) was collected via standardized questionnaires available on the AMAG (http://www.myopathie-atypique.be) and the RESPE (https://respe.net) websites. These forms were completed via email or phone contact with the owners or veterinarian whenever possible. Additionally, cases were also gathered by direct contact via mail or phone between owner/veterinarian and the principal investigator of this study. Information about the management and environment of diseased equines was obtained from the animals’ owners whereas clinical data was collected from veterinary surgeons. Cases occurring between the 1 September to the end of February were classified as “autumnal cases” and those from the 1 March up to the end of August as “spring cases”.

## 3. Results

### 3.1. Literature Review

Among 68 records identified in PubMed, five were rejected as they did not concern AM. From the search in Scopus, 514 results were obtained that were refined by selecting only research articles. From the 82 remaining studies, 63 were out of the scope and were discarded. In total, more than 127 documents (research articles and abstracts) were obtained.

### 3.2. Epidemiological Data

Epidemiologic data from 3039 cases were recorded in 14 different countries from autumn 2006 to 30 November 2019 (Figure 1, Table 1). This data set includes all cases communicated to the surveillance networks with a tentative diagnosis of AM. During this period, 14 autumnal outbreaks were encountered with a mean (± S.D.) of 164.6 ± 153.3 (median 79.5) reported cases. For spring, 12 outbreaks were recorded with a mean (± S.D.) of 56.5 ± 77.3 (median 24.0) diseased horses per outbreak. For all outbreaks together, the mean (± S.D.) number of cases is 112.5 ± 132.4 (median 41). Parts of these data (n = 824) have already been analyzed to define risk (Table 2) and protective (Table 3) factors on cases in Belgium [17], in the UK [12] and at a European level [11,15]. Equines particularly at risk for AM were found to be young (i.e., less than 3 years of age) and inactive animals with normal body condition score and receiving hay in autumn [17]. The risk of intoxication was also associated with full time pasturing in a humid environment where dead leaves pile up in autumn, with the presence of trees and/or woods and thus presumably exposed to the above-mentioned maple trees [12,15,17]. During the ten years that have passed since the last epidemiological study performed at an European level [11,15], 2433 new cases have been reported, reaching a total number of 3039 cases available for the current study. The whole database (i.e., 2006–2019) was cleaned by removing all equids that were not at pasture at the onset of clinical signs or within the week preceding these signs, equids that were diagnosed with another disease as well as equids having a low probability of intoxication (e.g., no pigmenturia and serum creatine kinase activities <10.000 IU/L; normal values 50–200: IU/L) according to van Galen et al., 2012 [11]. As opposed to the study of van Galen et al., 2012 [11], cases too poorly documented to make a definitive diagnosis have been retained in the study group. A total of 2371 cases was included in this study. The age distribution of these cases over the years are presented in Figure 2. The weekly occurrence of AM during the spring and autumnal seasons may be found in Figure 3a,b, respectively.

## 4. Discussion

The origin of the results consists of previous epidemiological studies modified and completed by the analysis of the newest data. The results contribute to answer the FAQs regarding horse feeding and management practices to reduce the risk of AM.

### 4.1. FAQ1: “Which Maples Are Toxic? Is this Tree a Maple and If So, Is It Toxic?”

The question about which maple trees are toxic is often associated with a request to identify trees on the pasture. The database consultation indicates that 99% of pastures contain or are directly bordered by tree. However, looking at the data from 2014 up to now, it is observed that 20% (92/456) of AM horse owners could not answer if seeds and/or seedlings of sycamore trees were present in their meadow. Despite the educational material available on the Internet (https://en.wikipedia.org), horse owners and veterinarians are still struggling to recognize the different maple species (personal observation). This phenomenon is accentuated due to the numerous erroneous descriptions in the literature [18]. A guide from Renaud et al., (2019) is available [19], helping the different actors to differentiate the three *Acer* species commonly found in European pastures where cases have been declared [20].

The maple genus includes approximately 561 species [21]. Some of them are extensively planted as ornamental trees because of their autumnal color. As a result, there is not only a demand to distinguish non-toxic trees (*Acer platanoides* (Norway maple) and *Acer campestre* (Field maple)) from *Acer pseudoplatanus,* but also many questions regarding the potential toxicity of other maple species. Even though up to now, not all maple trees have been tested, it is one of note that almost 50 years before the discovery of the cause of AM, the incriminated toxins had already been tested in many *Acer* species [3]. Among the tested species, the following species have tested positive for HGA and/or MCPG (non-exhaustive list): *Acer palmatum, Acer japonicum, Acer macrophyllum, Acer spicatum, Acer saccharinurn,* and *Acer saccharum*. These exotics species may be found in ornamental gardens and may spread to the neighboring regions [22]. Therefore, these *Acer* species might ultimately represent a risk of intoxication for equids.

### 4.2. FAQ2: “How Can AM Be Prevented (at Pasture Level)?”

Atypical myopathy occurs seasonally with outbreaks starting in autumn that may continue in early winter. On the contrary, spring outbreaks usually cease before summer. At pasture level, the risk can be decreased (1) by avoiding contact with toxic plant material and (2) by favoring low-risk meadows for pasturing during autumn and spring.

#### 4.2.1. Avoid Contact with Toxic Plant Materials

Former epidemiological studies identified access to dead leaves piled up in autumn, the presence of trees and dead wood on pastures as risk factors for AM [15,17]. This observation is presumably due to the presence of *Acer pseudoplatanus* and the ingestion of samaras and seedlings in autumn and spring, respectively [23,24,25]. It is therefore important to be able to recognize *Acer pseudoplatanus*, its samaras and seedlings. When in doubt, professional expertise should be sought to identify the tree (botanists and/or forestry agents might be of help). Recently, it has been suggested that flowers falling from sycamore trees after heavy rainfall and/or wind could be an additional source of intoxication [26].

Depending on weather conditions, samaras of *Acer* species may be able to spread their seeds up to several hundred meters [27]. Therefore, pasture contamination with seeds or seedlings is not necessarily linked to the presence of a tree on the pasture. In early autumn, especially after windy weather has dispersed sycamore samaras, it is recommended to equids’ owners to identify contaminated areas in their pasture. The removal of seeds may help to prevent AM [28]. However, when samaras are too abundant and/or too widely dispersed within the premise, grazing in the affected area must be prohibited. Another way to limit grazing to areas free of fallen seeds and/or flowers and/or seedlings is to create parcels within the pasture [26].

Additionally, the spreading of manure and/or harrowing of pastures was found to increase the risk of AM [17]. This practice might favor the dispersal of the toxic material throughout the pasture and subsequent intoxication of horses.

Regarding prevention at the pasture level, there is growing interest in the disposal of seedlings. It is worth noting that seedlings still contain HGA after herbicidal spraying or mowing [29]. These techniques are therefore ineffective regarding the destruction of toxic material.

#### 4.2.2. Use or Create Low-Risk Pastures

Permanent pasturing was found to be a risk factor. This is most probably due to the associated decrease in grass quantity, which leads equids to ingest the etiological agent [12,15,17]. Our database indicates that, in 64% of our cases, the pasture grass was bare or absent of grass. This observation correlates with previous epidemiological studies [11]. A good pasture management (for example pasture rotation) is advised in order to offer lush pastures. Indeed, a green meadow will limit the ingestion of seeds and seedling by horses allowing them to eat mainly grass [11,17].

Pastures particularly at risk for AM have *Acer pseudoplatanus* in their vicinity. Grazing on these meadows should be avoided during the risky seasons (see FAQ4). Furthermore, humid pastures are of particular risk for AM and grazing should therefore also be avoided on these pastures [17]. HGA is a water-soluble toxin that may pass from plants to water by direct contact [26,30]. This solubility might explain the risk associated with humidity and the protective factor linked to drinking water provided via the distribution network. For this reason, only pastures that do not contain rivers and/or freestanding water should be used during the risky seasons.

### 4.3. FAQ3: “How Can AM Be Prevented (at Horse Level)?”

All type of equids were affected by AM including donkeys (1.6% of the cases) and zebras (3 cases from zoological parks) with no highlighted risk factor associated with any species. The first epidemiological study highlighted that young horses, especially those <3 years of age were the primary affected group [17]. Later on, van Galen et al. (2012) [11] found that all age groups were represented. Our data *suggest* a gradual change in age distribution of cases over the years (Figure 2). In 2006, 71% of affected equids were less than 3 years old, whereas now this age group represents only 36% of individuals. This finding is unlikely to be explained by acquisition of immunity to the toxins since some survivors of AM did succumb to a second episode of the disease (unpublished data). An explanation could be that the population at risk is increasing because toxic pressure has increased over the years. The practical usefulness of this information is that all equids must be considered at risk, whatever their age. At the animal level, the risk can be decreased by (1) management and (2) feeding practices.

#### 4.3.1. Management of Grazing Time

The intoxication is intimately linked to pasturing as van Galen et al., (2012) reported that 98% of affected horses were at pasture at onset of clinical signs [11]. Our data confirms this information with 99.8% of horses being at pasture when clinical signs declared. The few remaining cases had been stabled for less than a week which implies these animals may have been in contact with the toxin before. Up to now, not a single case of our database has been confirmed in horses that had no access to pasture and/or paddock based on HGA and MCPA-carnitine detection (unpublished data). Thus, it is advised to stable horses during autumn and spring outbreaks if seeds or seedlings are, or may be, present at pasture [26]. However, keeping a horse in the stable day and night may be difficult and is not considered as good practice in animal welfare [31,32]. Interestingly, the limitation of grazing time to less than six hours a day was found to be a protective factor [15]. Consistent with a previous study of UK cases [12], 97.5% of equids of our database had spent more than six hours per day at pasture. This observation suggests that the length of exposure to the toxins appears to be a determining factor in the risk of AM.

Specific weather conditions have been linked with AM outbreaks [9,10,33,34] whereas weather-dependent pasturing time (i.e., stabling horses when inclement weather is forecast) reduces the risk of AM [17]. Reducing pasturing time according to weather conditions was not a recommendation implemented by owners in our cases since less than 1% of them were in compliance with this preventive measure. Our data reinforces the value of this preventive measure.

#### 4.3.2. Feed and Water Supply

Receiving supplementary feeds (hay, straw, complete mix, oats, barley, and/or corn) throughout the year decreases the risk of AM [17]. Atypical myopathy results from an energetic imbalance subsequent to HGA and MCPG poisoning. Feed provides energy substrates (especially carbohydrates) that supports the energetic metabolism and also vitamins and antioxidants known to increase the chance of survival [15]. The mitochondrial enzymes inhibited by HGA are flavin adenine dinucleotide (FAD) dependent. This cofactor originates from riboflavin (vitamin B2) suggesting that it would be useful to give this vitamin [35]. Alfalfa is an excellent natural source of riboflavin as well as, to a lesser extent, the hay from common grass [36]. In addition, well-nourished horses might be less tempted to ingest samaras and/or seedlings. Among horses receiving supplementary feeds in their daily diet (64%; n = 665), our data indicates that 61% received concentrated feed (complete mix, oats, and/or barley) and 50% had access to a salt block providing minerals. However, a salt block did not prevent these animals to be intoxicated.

Giving hay in autumn was identified as a risk factor [17] (and 40% of our cases receiving supplementary feed were fed with hay only). Indeed, hay may contain seeds and seedlings with detectable HGA concentration after several months [29] and even years of storage [30]. Gonzales et al. (2019) suggests that AM might occur in stabled horses [29] but this hypothesis is not sustained by our data since, as above-mentioned, not a single case has been confirmed by blood testing in equids with no access to pasture and/or paddock. However, we do agree that giving hay produced from contaminated pasture would increase the risk of AM in horses kept at pasture. It is probably wise not to produce hay/haylage and/or silage from pasture areas in the vicinity of sycamore trees [29]. Providing hay in autumn is controversial. Indeed, this practice was found to be a risk factor in a case-control study [17] but that appears nevertheless as a protective factor when comparing management practice in pasture of cases versus pastures of controls [15]. From this result, we can suggest that forages free from toxins should be given *at libitum,* but hay should neither be placed on the ground, nor under sycamore trees, since both practices could increase the risk of ingesting toxic material.

#### 4.3.3. Drinking Water

Water supplied by the distribution network [17] or stored in a tank or in an bathtub [15] are protective factors. These observations suggest that water from other sources may be contaminated by the toxins. This hypothesis is reinforced by the study of Renaud et al., (2019) [30], which showed that HGA is released by stagnant flowers or seeds from *Acer pseudoplatanus* in contact with water. On the other hand, when water is dripping off flowers, no HGA is detected in collected water. However, this latest observation can be modulated by the fact that (1) the concentration of HGA can be below the limit detection threshold of the quantification method or (2) that HGA might be degraded in water. To the authors knowledge, there is no published study about the stability of HGA in water.

### 4.4. FAQ4: “ Our Pasture Is Surrounded by Sycamore Maple Trees, but No Case of AM ever Occured in Our Grazing Horses. Does this Mean the Pasture Is Safe for Our Animals?”

Our data and former epidemiological studies indicate that unexplained sudden deaths of horse(s) had been noted on 22% of the pastures where cases were grazing [11]. In other words, 80% of cases were grazing in pastures that had no history of previous death of equid(s) (regardless of the suspected cause). Atypical myopathy is an emerging disease and a pasture surrounded by sycamore trees should not be considered as safe for pasturing horses.

### 4.5. FAQ5: “When Does the Risk of AM Start and Stop in Autumn and Spring?”

As previously reported, cases of AM occur more frequently in autumn (76%; n = 1801) than in spring (24%; n = 570). The expressions “autumn” and “spring” should not be taken *stricto senso*, since the autumnal clinical series are continuing in early winter and some spring cases are occurring after the 21 June. It is worth noting that 94% of “spring” cases occurred between the 1 March and the 31 May and 94% of “autumnal” cases occurred between the 1 October up to the 31 December (Figure 3a,b).

The cause of autumnal outbreaks cessation is not precisely known. Before discovering the etiology of AM, it was hypothesized that severe frost might destroy the etiological agent since outbreaks tend to cease after several days of deep freezing [10]. Now that HGA has been described as incriminated toxin, this hypothesis can be refuted since laboratory investigation showed that HGA is unaltered after several freeze–thaw cycles [37]. In our laboratory (unpublished data), we have collected samaras from the environment on a weekly basis since 2016 from now and, with very few exceptions, HGA has always been detected in seeds of sycamore tree. However, clinical series of AM usually fully stop in winter and resume with the germination of the samaras (personal observation). The analysis of HGA concentration over time in samaras fallen on the ground showed a high variability from tree to tree and from week to week thus impeding an easy interpretation of the evolution of toxicity. These field studies were complicated by the fact that the samaras collected on the ground had fallen at very different times. Therefore, the cause of the ceasing “autumnal” outbreaks in winter is not known but could result from a reduction in accessibility (e.g., adheration to the ground following rain and frost) and/or a decrease in toxins’ concentration).

Regarding spring outbreaks, horse owners wonder if the case series stops because the seedlings have lost in toxicity. Actually, the end of spring outbreaks may not be explained by the disappearance of the toxicity since the seedlings remain toxic [26]. It is however hypothesized that spring outbreaks cease following a relative decrease in risk of intoxication by grazing. This reduced risk of intoxication might result from (1) a lusher meadow, (2) the observed decrease in toxicity of seedlings with their growth [26], (3) a decrease in palatability of older plants [26], (4) less frequently encountered weather conditions favoring toxicity and (5), a significant natural disappearance of seedlings. Regarding the latter, only a small percentage (<20%) of seedlings recorded in early spring on heavily contaminated pastures are still present in early summer (unpublished data). This observation added to the fact that herbicidal spraying do not reduce HGA concentration in sycamore seedlings [29] questions the benefit of herbicide treatments.

## 5. Conclusions

As there is no specific treatment for AM yet, prevention is the key. The risk of developing AM results from the combinations of protective and risk factors. In order to reduce the risk of AM, it is advised to avoid humid pastures, permanent pasturing, spreading of manure, and contact with sycamore plant material. During the risky periods pasturing time should be modulated according to weather conditions and limited to less than six hours a day. Grazing equids should receive supplementary feeds, with preferences for feeds containing riboflavin. When hay or silage are fed, it is necessary to exercise caution ensuring the forages are toxin-free. Also, it is advised to supply a salt block and provide drinking water from the distribution network. It is worth noting that AM is an emerging disease and equids of any age and all pastures with a sycamore tree in the vicinity must be considered at risk. These preventive measures should be implemented for a period of 3 months twice yearly, starting in March for “spring cases” then again in October to prevent “autumnal cases”. As mentioned before, these are the critical seasons and samaras or seedlings are likely to be present on the pasture. A French version of this paper with additional illustrations can be found in Appendix A. 

## Figures and Tables

**Figure 1 animals-10-00365-f001:**
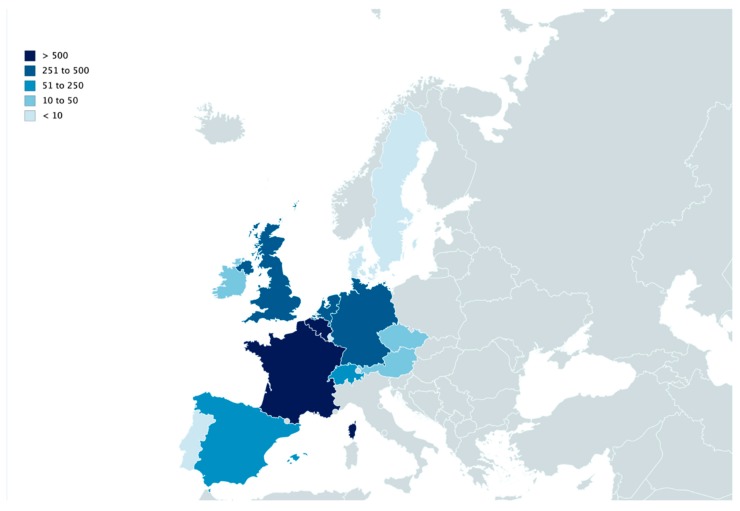
European distribution of atypical myopathy cases notified to the disease surveillance networks from autumn 2006 to November 2019.

**Figure 2 animals-10-00365-f002:**
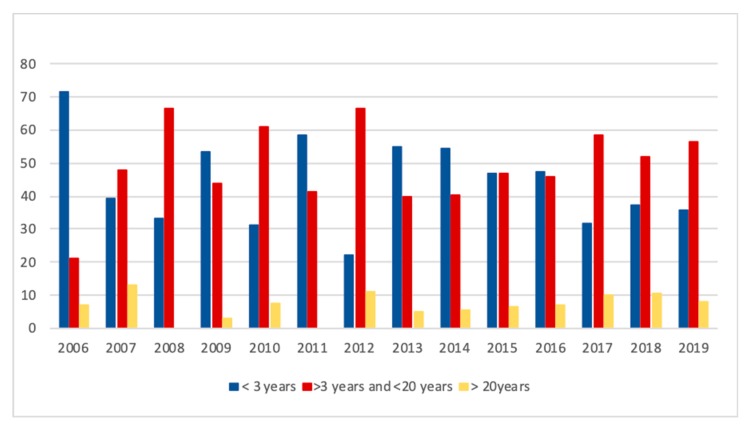
Frequency distribution of equids with age categories: <3 years, >3 years and <20 years and >20 years old (n = 1510) over the study period (2006–2019).

**Figure 3 animals-10-00365-f003:**
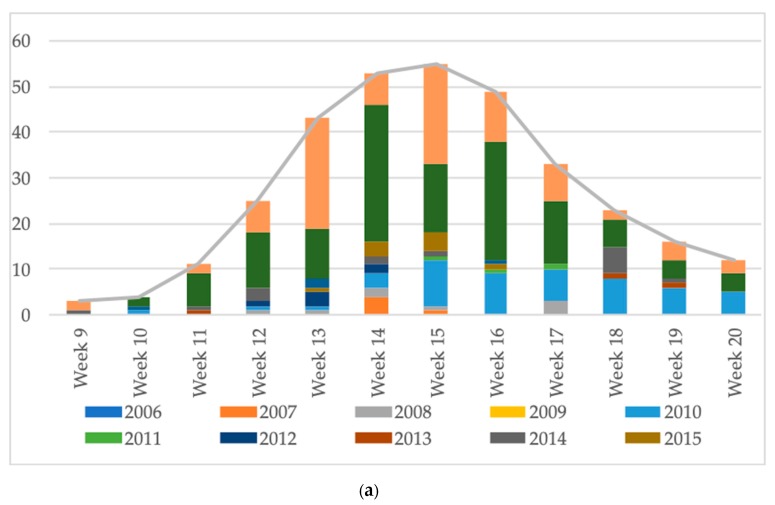
(**a**). Spring cases: weekly occurrence of atypical myopathy cases from week 9 (1 March) up to week 20 (31 May) over the study period (2006–2019); (**b**). Autumnal cases: weekly occurrence of atypical myopathy cases from week 41 (1 October) up to week 52 (31 December) over the study period (2006–2019).

**Table 1 animals-10-00365-t001:** Total number of atypical myopathy cases in Europe notified to the disease surveillance networks from autumn 2006 to November 2019 (n = 3039).

Year	2006	2007		2008		2009		2010		2011		2012		2013		2014 *		2015		2016		2017		2018		2019		Total/Country
Countries	Autumn	Spring	Autumn	Spring	Autumn	Spring	Autumn	Spring	Autumn	Spring	Autumn	Spring	Autumn	Spring	Autumn	Spring	Autumn	Spring	Autumn	Spring	Autumn	Spring	Autumn	Spring	Autumn	Spring	Autumn	
Austria	0	0	0	0	0	0	0	0	0	0	4	0	0	0	0	0	0	0	0	0	0	0	0	4	9	0	0	**17**
Belgium	46	7	18	0	5	0	69	13	14	3	18	0	1	0	141	8	51	5	6	8	52	20	5	1	191	49	1	**732**
Czech Republic	0	0	0	0	0	0	0	0	0	0	0	0	0	0	9	0	0	0	0	0	2	17	0	0	19	2	0	**49**
Denmark	2	0	0	0	0	0	2	0	0	0	0	0	0	0	0	0	0	0	2	0	0	0	0	0	2	0	0	**8**
France	32	1	11	18	11	0	134	106	32	10	40	16	4	1	64	13	71	9	24	8	194	181	31	18	114	47	26	**1216**
Germany	7	0	3	5	0	0	93	21	2	2	59	2	0	0	24	1	8	0	0	0	21	20	4	0	33	4	2	**311**
Ireland	0	0	0	0	0	0	2	0	0	0	0	0	0	0	2	0	38	0	0	0	1	2	0	0	1	0	0	**46**
Luxembourg	1	0	0	0	0	0	2	0	0	0	0	0	0	0	0	0	0	0	0	0	0	0	0	0	0	0	0	**3**
Portugal	0	0	0	0	0	0	0	0	0	1	0	0	0	0	0	0	0	0	0	0	1	0	0	0	0	0	0	**2**
Spain	0	0	0	0	0	0	0	0	0	0	52	0	0	0	0	0	0	0	0	0	0	0	0	0	0	0	0	**52**
Sweden	0	0	0	0	0	0	0	1	0	0	1	0	0	0	0	5	0	0	0	0	0	0	0	0	0	0	0	**7**
Switzerland	0	0	9	0	0	0	31	3	0	0	6	0	0	1	12	0	0	0	0	0	9	3	0	0	7	0	0	**81**
The Netherlands	13	0	3	0	2	0	34	7	0	0	0	0	0	0	18	1	4	0	0	0	1	2	0	2	9	10	1	**107**
United Kingdom	1	0	13	0	0	0	39	20	3	0	33	6	2	2	52	13	154	20	2	1	11	11	3	1	13	3	5	**408**
**Total/season**	**102**	**8**	**57**	**23**	**18**	**0**	**406**	**171**	**51**	**16**	**213**	**24**	**7**	**4**	**322**	**41**	**326**	**34**	**34**	**17**	**292**	**256**	**43**	**26**	**398**	**115**	**35**	**3039**
**Total/year**	**102**	**65**		**41**		**406**		**222**		**229**		**31**		**326**		**367**		**68**		**309**		**299**		**424**		**150**		

Comments: This last counting replaces all previously published data; the word “autumn” should not be taken strictly since clinical series continue into early winter; (*) in 2014, atypical myopathy was linked to *Acer pseudoplatanus* (sycamore maple) in Europe.

**Table 2 animals-10-00365-t002:** Risk factors identified in from former epidemiological studies [12,15,17].

Category	Risk Factors	Odds Ratio	95% Cl for Odds Ratio
**Demographic data**	Young horses (<3 years)		
	Thin (or normal weight *)	3.08 [17] (b) (3.85 [15]/2.20 [17] (b))	1.01–9.39 [17] (b) (1.77–8.37 [15]/1.01–4.79 [17] (b))
**Horse management**	At pasture 24 h a day all year round	5.42 [15] 3.07 [17] (b) in winter 3.78 [17] (b) in spring 23.2 [17] (b) in summer 10.9 [17] (b) in autumn	2.577–11.42 [15] 1.45–6.50 [17] (b) in winter 1.49–9.59 [17]] (b) in spring 1.41–382 [17] (b) in summer 3.56–33.4 [17]] (b) in autumn
	Not physically active	11.8 [17] (b)	5.02–27.8 [17] (b)
**Feeding practice**	Hay given in autumn	4.09 [17]] (a)	1.18–14.1 [17] (a)
**Pasture**	History of previous death of horse(s) on the pasture	4.45 [17]] (b)	1.61–12.29 [17] (b)
	Lush pasture in winter	3.95 [17] (b)	1.49–10.46 [17] (b)
	Sloping pasture/steep slope	3.43 [15] 3.70 [17] (b)	1.52–7.77 [15] 1.58–8.68 [17] (b)
	Access to dead leaves piled up in autumn	11.11 [15] 10.47 [17] (b)	4.82–25.59 [15] 2.82–40.88 [17] (b)
	Presence of trees at pasture *	7.82 [15]	1.99–30.73 [15]
	Dead wood at pasture	3.12 [15]	1.42–6.84 [15]
	Humid pasture	2.63 [17] (b)	1.29–5.36 [17] (b)
	Pasture surrounded by or containing a stream/river	2.78 [17] (b)	1.24–6.19 [17] (b)
	Spreading of manure	5.73 [17] (b)	2.40–13.69 [17] (b)

(*) Parameters that are not consistent among studies [15] cases with a high probability of or confirmed atypical myopathy vs. and the cases with a low probability of having AM or with another diagnosis [17] (a) confirmed cases vs. clinically healthy co-grazing equidae [17] (b) confirmed cases vs. control horses.

**Table 3 animals-10-00365-t003:** Protective factors identified in former epidemiological studies [12,15,17].

Category	Protective Factors	Odds Ratio	95% Cl for Odds Ratio
**Demographic data**	Overweight	0.25 [17] (b)	0.09–0.69 [17] (b)
**Horse management**	Frequent deworming	0.11 [17] (a) 0.05 [17] (b)	0.01–0.67 [17] (a) 0.01–0.16 [17] (b)
	Regular vaccination	0.10 [17] (b)	0.05–0.21 [17] (b)
	Regular physical activity	0.08 [17]	0.03–0.19 [17]
	Weather-dependent pasturing time in spring and in autumn	0.24 spring [17] 0.10 autumn [17]	0.06–0.89 spring [17] 0.02–0.56 autumn [17]
	<6 h at pasture per day	0.04 [15] 0.62 [17]	0.01–0.19 [15] 0.16–2.36 [17]
	No access to pasture	0.03 [15]	0.00–0.22 [15]
**Feeding practice** **and water supply**	Supplementary feeds all year round	0.17 [15]	0.05–0.59 [15]
	Silage and concentrate feed in autumn + corn in winter	0.20 [17] (a) for silage 0.19 [17] (a) for concentrate feed 0.22 [17] (a) for corn	0.04–0.94 [17] (a) for silage 0.04–0.87 [17] (a) for concentrate food 0.05–0.93 [17] (a) for corn
	Salt block (all year)	3.52 [12] 0.20 [17]	1.08–11.47 [12] 0.09–0.40 [17]
	Water provision in tank/bathtub	0.25 [15]	0.09–0.69 [15]
	Water supplied by the distribution network	0.39 [17] (b)	0.17–0.88 [17] (b)
**Pasture**	Gentle slope	0.34 [17] (b)	0.14–0.84 [17] (b)

[12] Survivors vs. nonsurvivors [15] cases with a high probability of or confirmed atypical myopathy vs. the cases with a low probability of having AM or with another diagnosis [17] (a), confirmed cases vs. clinically healthy co-grazing equidae [17] (b), confirmed cases vs. control horses.

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
