# Peer review of "Answers to the Frequently Asked Questions Regarding Horse Feeding and Management Practices to Reduce the Risk of Atypical Myopathy"

_animals, 2020, doi:10.3390/ani10020365_

Round 1

Reviewer 1 Report

This study gives important and deepened knowledge and practical advice on atypical equine myopathy caused by intoxication from Acer spp. This is an emerging and deadly disease, that has killed thousands of European horses. It is clear from their investigations that there is a great need of knowledge transfer to the horse industry to protect more horse’s lives. The manuscript has been considerably improved since the first version. However, there are some major concerns mainly in questions regarding clarity and logic. A revision is suggested.

Main suggestions:

The manuscript does not follow expected logic which causes confusion. Rearrangement is suggested, so that the main objective of the study comes first through Introduction, Materials, Methods, Results and Discussion, and that secondary aims follow in decreasing and logical order. Further, that all results are moved to results section, and not introduced in the Discussion.

It is perceived that the “FAQ’s” in fact are = the main research questions, as stated in the headline of the paper, and (a little bit poorly defined) in the Introduction. So, the suggestion is to present the FAQs in the Introduction as (principal) Aims, use Literature review and Epidemiological survey as Methods and to give the results of the investigation (answers of the FAQ’s) in Results. These results do not come now until in the Discussion – which is unusual and does not seem logic.

Please indicate somewhere how many outbreaks /locations that are represented in the study and the median number of sick horses per outbreak, with an interval.

Please describe the statistical methods used for all the analyses and the level of significance used.

Lines 40-41: Are “humid pastures, permanent pasturing and spreading of manure” really risk factors to avoid if there are no sycamore maple trees around? Is this a relevant advice, generally, anywhere in Europe? Otherwise, please re-write so that inclusion of sycamore maple trees is a pre-requisite for these three other factors to be risk factors.

Line 42 – Same as above, “reduce pasturing time”- add: “for horses on risk pastures”. If you don’t see a need to generally advise to reduce pasturing time also when there are no sycamore maple trees in the vicinity.

Lines 83-86 – Please clarify how the number 74% is achieved, is it a median or average of a certain population? The word “generalization” is unclear- I suggest changing for a clearer explanation, that could be traced to its original data and compared to future studies.

Lines 87-90 – Please comment, was there not a multifactorial analysis to test that hypothesis of the combined factors level of initial clinical signs, place for care and treatment provided, and how they affect the survival rate?

Lines 167-168 – Please clarify the definition of “young horse” (definitions could be placed in Materials & methods), and explain the conclusion of the significantly increased risk for young horses with numbers. Could not find this in Tables 1-2. Was every one of the described 3039/2731 cases young? Are all horses in the tables “young horses”? Unclear. Is it possible to provide a risk factor or percentage compared to other age groups, etc.

Please clarify why only “normal body condition” is mentioned here but in Table 2, it seems that “thin” is also a risk factor.

Please clarify why “receiving only hay in autumn” is mentioned here, but in Table 2, the wording is “Hay given in autumn”. It is confusing, is the correct interpretation that “hay” or “hay only” is a risk factor? I suggest including an explanation of this factor.

Lines 168-169 –Please clarify, what is defined as feed supplementation, and was feed supplementation never given to any horse at all of the 3039/2731 case horses, or not in a majority of cases (%)?

Line 208 – There is a lot of data in the tables and maps, and it would be helpful if the results were summarized in the text. Such as: which season is worse, how many cases were found in autumn and spring respectively? Any difference between countries in this? Which countries reported most cases? Have all countries reported a steady number of cases during the period or are there large differences between years?

Line 213The table is cut in the right-hand side of the provided pdf-file after the column for year 2017, so it was not possible to review fully. Therefore, it was for example not possible to see the total sum of the cases per country, and the total number of the included cases. Suggest including more information in the title of the table, such as: “Table 1. Reported atypical myopathy equine cases (n=2371??) per country, season and year from 2006 to 2019 in Europe “

Line 223Table 2 and 3 are very interesting but somewhat difficult to comprehend, please help the reader. Are these positive and negative variables that are significantly coupled to diseased horses, as compared to control horses (at the same premises?), please comment whether this is a possible interpretation, or how the table shall be interpreted for the reader that is less experienced in epidemiology, such as a practicing veterinarian or interested owner. It is partly indicated in small text under the tables but warrants a better explanation in the running text for clarity.

What is the study population (number of outbreaks and number of case and control horses), is it different for each variable? Please indicate in table titles or per variable, or elsewhere.

How come one variable is named “At pasture 24h a day all year round” but then some of the results refer to 24 h pasture only part time of the year? Could this variable be called “At pasture 24h a day”?

It would be helpful if the different seasonal pasture patterns somehow were coupled to and presented by at what time of year the horse was ill. For example, is it interpreted of the same relevance, if the horse was ill at the same time as it was kept outdoors 24/7, or during a completely different time of the year?

Same question for the variable “Access to dead leaves piled up in autumn” – is this a relevant question for horses sick in spring, or should this variable rather be reported /analyzed grouped into horses sick in autumn and spring separatedly? Or are the leaves still there in spring? Unclear.

“Hay given in autumn” -see comment for line 167-168, plus the comment above if this question should be analyzed separately for spring and autumn cases?

“History of previous death of horse(s) on the pasture” – what is the definition of previous? Same year? All years? How long back? Especially: Please comment if other cases during the same outbreak (same year) were included here, so that these numbers might be biased/confounded by horses that were not index cases, but within the same outbreaks as others in the study population? Maybe this question is better analyzed per outbreak rather than per horse?

Line 224, Table 3 – Salt block (all year) – please explain the two different results

Discussion – Please discuss whether the results of this study can be extrapolated to the whole European or global horse population regarding risk factors and responses to FAQs for this disease, or if there are any limitations. Discuss any bias as to the distribution of countries.

As already mentioned, because giving responses to FAQs is perceived as the aim of this study, the responses are traditionally expected to be found in the Results, and the discussion of these results in the Discussion. (If this basic concept is changed, please explain the concept to the reader. One way would be to call it Results & Discussion if it is allowed by the journal).

My suggestion is to provide clear correct answers easily found directly after each question (FAQ1-5) in the Results section. Sometimes these answers are lacking in the current manuscript, e, g. FAQ 1, and sometimes they are there, but not clearly found under the question, e. g. FAQ2, FAQ3. That would imply moving most of the FAQ-text,  the parts that are the clear answers to the posed questions, to the Results.

Then return to the FAQs in Discussion with the valuable discussion on how these issues can be interpreted, identified challenges, future suggestions etc (as is often very well discussed already).

It would be helpful to elucidate/discuss if “permanent pasture”, “humid pastures”, access to natural water, times on pasture, weather dependent pasturing etc sources constitute risk factors even in the absence of sycamore maple trees. You should be able to analyze the data this way if you have access to responses with and with without such trees. If you have responses only from farms with sycamore maples, you must be clear and state to the discussion that these variables are only valid to extrapolate to horse populations located with sycamore maples trees.

Lines 677-682 – Again, are all these advices general to all horse populations or only those residing near sycamore maples trees? Please clarify.

Minor comments:

Line 26 – spelling: piling

Lines 44-46 – Suggest clarification, change to: “94% of cases occur over two 3-month periods, starting in October and in March, for cases resulting from seeds and seedlings ingestion, respectively”

Line 65 – Suggest clarification: “the ingestion of some/certain species of maple (Acer)”

Line 79 – “not-exercise related” or “not-exercise-related” or “an acute rhabdomyolysis syndrome unrelated to exercise”, unsure which one is most correct

Line 83 – “in more than 50% of cases” does it relate to all the signs or the last sign, unclear?

Line 86 – Please add what the number is compared to (mortality rate), and if it is a significant difference.

Line 91 – Suggest introducing the terms “passive disease surveillance” or “syndrome surveillance” and “risk factor studies” for the work that AMAG is/was doing

Line 153 – change “with” for “and”: “between owner/veterinarian and the principal investigator”

Lines 199-201 – Are these factors significantly associated with disease?

Lines 200-202 – Suggest rewording: “During the ten years that has passed since the last European epidemiological study of cases from 2006-2009 [11,14], 2433 new cases were noted, providing a total population of 3039 reported cases available for the current study”

Lines 202-208 – Two reasons for exclusions are mentioned. How many cases were excluded to each of these?

Line 341 + Figure 2 – This is a not earlier introduced result and should not be presented in the Discussion, move to Results.

Line 543 – Suggest discussing this finding. It would be interesting to hear your theory on why shorter turnout would be protective over longer turnout times. Is it not possible/easy to ingest the necessary amount of toxin to become intoxicated in <6 hours grazing? What does the literature say about pharmacokinetics and metabolizing of different doses of the toxins?

Line 553 – Is “ beverage” the correct term for horses drinking water supply?

Line 556 – Omit double word: “hay hay”. What do you mean by “hay (generally speaking)“? Do you mean forage?

Lines 566-568 – Do you mean: “Indeed, hay may contain seeds and seedlings with detectable HGA concentrations even after several months [27] and years of storage [28].”?

Lines 573-575 – Please clarify. This sentence is difficult to understand and lacks references: “Indeed, this practice was found to be a risk factor in a case-control study but that appears nevertheless as a protective factor when comparing management practice in pasture of cases versus pastures of controls.”

Lines 622-623 – References are lacking.

Lines 623-625 – Check spelling “autumnal” (also in line 639). Please provide exact ranges of index case dates for inclusion into “spring” and “autumn” groups, suggestedly in the Material and Methods section.

Lines 625-627 and Table 3a and b – Another examples of new data that should be introduced in Results, not in Discussion. But there are many more such examples, suggest an overall check.

Lines 632-634 – What is “basis samaras”? Check the sentence, difficult to understand. During which period of the year was this performed?

Lines 634-635 – It would be interesting to know whether or how many of the outbreaks that showed cases both in spring and following autumn, or autumn and following spring (in Results).

Line 638 – Check spelling: samares?

Lines 645-646 – Do you mean ”relative decrease in risk for intoxication by grazing”?

Line 648 - Do you mean “encountered”? What are weather conditions favoring toxicity, references?

Line 651 – Unclear what you mean by “This observation” , if it is the disappearance of seedlings, it makes no sense with the rest of the sentence? Please clarify the reasoning.

Lines 682-684 – It is unclear why you imply that all types of horses are at risk because AM is an emerging disease. Suggest removing the word “thus”.

Lines 684-685: Suggest changing to: These preventive measures should be implemented for a period of three months twice yearly, starting in March to prevent “spring cases” and then again in October to prevent “fall cases”.

Reviewer 2 Report

Dear Authors and Editors, I find the revised version of the manuscript improved an suitable for publication with some minor chnages:

table and figure layout and placement of tables, figures, table headings and figure legends needs to be checked, the current format with all track changes visible makes it problematic to see the final layout. In table 3, for salt block the figure says 3,52 but should be 3.52 The correct English term is concentrate feed (or just concentrates), not concentrated feed. suggest rephrase the sentence in line 554-555: "Atypical myopathy results from an energetic imbalance." as it would mean all horses in energetic imbalance (too high or too low) would risk AM, irrespectiv of if they have ingested the toxin or not. I still think figure 2 should include horses aged 3-20 years for enhanced clarity Reference style does not seem to follow journal guidelines.

Author Response

Dear Reviewer,

Thank you for your comments. Please find below a point-by-point response to your comments.

1) Table and figure layout and placement of tables, figures, table headings and figure legends need to be checked, the current format with all track changes visible makes it problematic to see the final layout.

The track changes have been removed for a better reading of the tables

2) In table 3, for salt block the figure says 3,52 but should be 3.52

Done according to your suggestion.

3) The correct English term is concentrate feed (or just concentrates), not concentrated feed.

Done according to your suggestion.

4) Rephrase the sentence in line 554-555: "Atypical myopathy results from an energetic imbalance." as it would mean all horses in energetic imbalance (too high or too low) would risk AM, irrespective of if they have ingested the toxin or not.

The sentence is replaced by “Atypical myopathy results from an energetic imbalance subsequent to HGA and MCPA poisoning.” (line 268)

5) I still think figure 2 should include horses aged 3-20 years for enhanced clarity

Done according to your suggestion.

6) Reference style does not seem to follow journal guidelines.

Changes done according to journal guidelines.

This manuscript is a resubmission of an earlier submission. The following is a list of the peer review reports and author responses from that submission.

Round 1

Reviewer 1 Report

This article is a review aiming to provide responses to common questions regarding management of horses to reduce the risk of atypical myopathy. The aim is good; there is a need to spread the knowledge acquired by research, reports and observations gathered during the last years on this serious and emerging equine disease. The facts included are probably correct, but they need a better presentation.

The main problem with the article is that it is poorly written and does not achieve the anticipated quality. Major revision of the English language, grammar and spelling and the way to present the arguments is necessary.

Introduction: it would be helpful to provide the interval of case mortality rates observed in farms, if 74% is not a reliable number.

The FAQs are not proper full questions.

It would help if the risk factors in Table 2 were somewhat better described, e.g., are they representing different levels of risk? 

The paper needs better logic and stringency, especially the discussion is too wordy, and the "responses" to the FAQs in the discussion need to follow a better line.

There were so many paragraphs, sentences and words that need linguistic review and would benefit from revision that it is too much to relate details here. I will be happy to review it again after a major revision and editing. 

Reviewer 2 Report

Dear Authors, Thank You for providing a comprehensive overview of this important subject. My first reaction on the format of the paper was that maybe it was not so good to structure the paper in this way, but as I read on I liked it a lot, so even though it may be unconventional, I think the format is fine for thism type of paper. I have some minor comments and questions (please see detailed list below), but on the whole, I think the paper is suitable for publication.

Detailed list of comments/questions:

Line 69-70: Here the language/grammar format suggests all readers knows that mortality rate of AM is 74 %, which we of course do not. Could this sentence be formulated in another way?

Line 90: frequent should be frequently

Line 106: Answer should be Answers

Line 109: The word "led on" seems to be used a little bit unconventionally here, perhaps replace with "performed using"

Line 100: meeting should be meetings

Table 1 is very difficyult to read due to its small size. Could it be placed in landscape layout?

Line 150-153: are these comments connected directly to Table 1? It is not really clear if they were supposed to be footnotes to the table or not. This would need to be revised.

Figure 1: in black/white print it is very difficult to identify the colour code

Table 2 and 3: would it be possible to include the OR and confidence intervals for the risk factors and protective factors? It would be of interest to provide information on how these factors/variables rank, e.g. is one variable more or less protective/risky than another?

In Table 3, The three last rows seems to be left-aligned and not centered

Line 174: a dot is missing before Despite

Line 179: incudes should be includes

Line 188: This should be These

Line 209: should be "...., it is recommended to owners of equids to identify..."

Line 212-213: the format of this sentence is different from other sentences in the paper, as it is a direct urge to horse owners, it seems a little malplaced. If it could be changed to a format more in alignment with the surrounding text it would not "stand out" as it does now.

Line 220: suggest add "horses" (or equids) after encouraging

Line 221: suggest replace "..., their pasture grass was bare or absent as found in..." with the following: "..., their pasture was bare or absent of grass as found in..."

Line 222: Here lush pastures are advised as a protecctive facor, but in Table 2 it is a risk factor at winter. This would need to be explained a bit more clearly.

Line 223-224: suggest replace last words of the sentence with: "...allowing them to eat mainly grass."

Line 219-224: was there any information about grazing pressure or stocking rate in the database ? As the risk of ingestion of unsuitable plants generally increase when pasture grasses are low in abundance it may add to preventive measures if a stocking rate can be calculated ?

Line 227: particular instead of particularly

Line 228-230: could it also reflect a difference in vegetation in different pasture land types? Humid pastures have other grass types than more dry pasture lands, which could affect the preference of which plants the horses are selecting for grazing.

Line 229: Drinking water could also come from a farm-individual water well (very common in many parts of Northern Europe) and not from the municipial distribution network. As I understand it, the main idea is to avoid horses drinking from rivers/streams, water puddles, lakes etc where Acer pseudoplatanus is present?

Line 234: double-wyping of "horses", (should be young horses ?)

Line 236: to avoid interpretation mistakes, it should be clarified -was represented in what?

Line 240-241: increased toxic pressure due to....? Increased presence of Acer pseudoplatanus in pastures?

Figure 2: I can´t help wondering where horses aged 3-20 would be placed in the figure? Especially considering the information given in lines 235-238. Also, when printed in black/white, it is not possible to identify which bar is which.

Line 252: animals, not animaly

Line 260-261: as in increased time of exposure to toxins due to increased time on pasture where grass is absent and the risk of ingesting Acer plants or seeds increases with increased hunger in the horses? Or do You mean something else?

Line 263: weather-dependent pasturing time would need to be explained as it is not clear what is meant by this?

Line 264: recommendation not recommandation

Line 264: "less than", not "than less"

Line 267: Animal food is termed feed (suggest change to "Feed and water"). Use "feed" consistently instead of "food".

Line 270: known, not know

Line 273-274: terms used here are not really correct: alfalfa is a forage plant. Hay is one way to conserve forages. So, the comparison should then be between alfalfa and grasses, not between alfalfa and hay. Reference needed for the statement of alfalfa as a plant rich in riboflavin.

Line 276: indicated not indicates

Line 277: I think a dot and a new sentence is required after "salt block", the sentence is very confusing as it stands now. 

Line 279: end of parenthesis missing

Line 280: what type of hay? grass hay? alfalfa hay? other? was it only hay and not silage or haylage?

Line 285: don´t you mean to horses kept in stable, instead of at pasture?

Line 286: produce not product

Line 287-289: not really clear what is meant here? You would need to clarify this.

Line 315: December, not Decembrer

Line 323-329: just a question: is there any similarity with e.g. oak acorns, which are more poisonous when green compared to brown independent of at which season or month they have fallen to the ground?

Line 331 and 336: disappearance not disapearance

Line 333: the term "lush" is actually not referring to anything other than how a grass is percieved by a human. I would guess that a higher abundance or availability of pasture grass is what was really meant here ? Or increased growth rate of pasture grass.

Line 334: add "in" before toxicity

Line 347: suggest rewrite as follows: "...treatment for AM, and AM has a high mortality...."

Line 348: sugges using "risk" instead of "chance" here as it is a serious disease

Line 348: omit "the"

References number 5 and 18 lacks sufficient information on where/by whom they have been published or will be published.
